# A systematic *in vitro* study of the effect of normoglycaemic and hyperglycaemic conditions on the biochemical and cellular interactions of clinically-available wound dressings with different physicochemical properties

Shirin Saberianpour[1,2], Gianluca Melotto[1,3], Rachel Forss[1,3], Lucy Redhead[1,3], Susan Sandeman[1,2], Nadia Terrazzini[1,2], Dipak Sarker[1,2], Matteo Santin[1,2]*

1 Centre for Regenerative Medicine and Devices, University of Brighton, Brighton, United Kingdom, 2 School of Applied Sciences, Brighton, United Kingdom, 3 School of Sport and Health Sciences, University of Brighton, Brighton, United Kingdom

* m.santin@brighton.ac.uk

**Data Availability Statement:** All relevant data are available at the University of Brighton Open

## Abstract

Diabetic foot, leg ulcers and decubitus ulcers affect millions of individuals worldwide leading to poor quality of life, pain and in several cases to limb amputations. Despite the global dimension of this clinical problem, limited progress has been made in developing more efficacious wound dressings, the design of which currently focusses on wound protection and control of its exudate volume. The present in vitro study systematically analysed seven types of clinically-available wound dressings made of different biomaterial composition and engineering. Their physicochemical properties were analysed by infrared spectroscopy, swelling and evaporation tests and variable pressure scanning electron microscopy. These properties were linked to the interactions with inflammatory cells in simulated normoglycaemic and hyperglycaemic conditions. It was observed that the swelling behaviour and evaporation prevention at different glucose levels depended more on the engineering of the fibres than on the hydrophilicity and hydrophobicity of their biomaterials. Likewise, the data show that the engineering of the dressings as either non-woven or woven or knitted fibres seems to determine the swelling behaviour and interactions with inflammatory cells more than their polymer composition. Dressings presenting absorbent layers made of synthetic, non-woven fibres supported the adhesion of monocytes macrophages and stimulate the release of factors known to play a role in the chronic inflammation. Non-woven absorbent layers based on carboxymethyl cellulose mainly stimulating the iNOS, an enzyme producing free radicals; in the case of Kerracel this was combined with a swelling of fibres preventing the penetration of cells. Kaltostat, an alginate-based wound dressing, showed the higher level of swelling and supporte the adhesion of inflammatory cells with limited activation. Knitted dressings showed a limited adhesion of inflammatory cells. In conclusion, this work offers insights

Repository (DOI: https://doi.org/10.17033/DATA.
00000321).

**Funding:** This work was supported by the UKRI
Engineering and Physical Sciences Research
Council (Grant No. EP/W023164/1). The funders
had no role in study design, data collection and
analysis, decision to publish, or preparation of the
manuscript.

**Competing interests:** The authors have declared
that no competing interests exist.

about the interactions of these wound dressings with inflammatory cells upon exudate
changes thus providing further criteria of choice to clinicians.

## 1. Introduction

Wound dressings are the medical devices of choice for the treatment of both acute wounds
(e.g. burns and surgical wounds) and chronic wounds that include diabetic foot, leg and decu-
bitus ulcers [1]. Each of these types of wounds affect millions of individuals worldwide [2, 3].
Acute wounds are expected to heal within two to four weeks unless complications such as
infections occur, whereas chronic wounds can remain unresolved for months, years or for the
whole patient's life often leading to poor quality of life, pain and, in several cases, to limb
amputations [4]. In all these cases, wound dressings are applied by the practitioners with the
intent of preventing mechanical and infectious insults as well as of keeping the wound moist, a
condition widely accepted to be necessary to healing [3, 5].

During the 1960s, it was acknowledged that gauze and other absorbent materials, such as
cotton and gamgee, were inactive products that obstructed the wound without promoting
wound healing. [6]. Consequently, a basic set of standards for "optimal dressing" was estab-
lished whereby the device would facilitate the ideal healing process of the wound in any given
clinical scenario [7]. In 1979, Turner delineated the performance parameters that constitute an
"optimal wound dressing". These parameters have more recently defined by the UK National
Institute for Health and Care Excellence (NICE) in their British National Formulary (BNF) as
able *"to ensure that the wound remains: moist with exudate, but not macerated; free of clinical
infection and excessive slough; free of toxic chemicals, particles or fibres; at the optimum temper-
ature for healing; undisturbed by the need for frequent changes; at the optimum pH value"* [8].

In recent years, various biomaterials have been utilized and engineered into wound dress-
ings that possess distinct compositions and structures to offer advantages in terms of water
retention, biocompatibility and therapeutic properties in different types of wounds [9]. Typi-
cally, a film made of silicone or polyurethane or poly(ethylene terephthalate) is used to protect
the wound from contamination and limit water evaporation. This layer is combined with one
or more absorbent layers made of either synthetic or natural polymers in the form of natural
or synthetic hydrogels or hydrocolloids (e.g. carboxymethyl cellulose, alginate, polyester). In
other types of devices, fibres are engineered in the form of low-adherence meshes mainly pro-
viding mechanical protection. This wide range of medical devices enable clinicians to choose
the dressings on the bases of the wound conditions; semi-permeable film and foam dressing
are used for wounds with low volumes of exudates, hydrogel and hydrocolloid dressings are
mainly used for wounds with moderate to relatively large volume exudates, and alginate dress-
ings are recommended for wounds presenting both large exudate volume and bleeding [1].
Furthermore, diverse technologies can be employed during wound dressing production to
incorporate active ingredients such as anti-bacterial silver nanoparticles or growth factors; the
use of the latter being very limited because of their relatively high costs and not sufficiently
proven efficacy [10]. In the case of hydrocolloids, the gelling process is accompanied by the
absorption of wound exudate into the fiber structure and as the fibers swell, the capillary struc-
ture in the nonwoven wound dressing is closed, thereby blocking the lateral spreading of liquid
[11]. The choice of the suitable dressing is also made wider to clinicians by the suitability of
most of these devices to be applied in combination as primary and secondary dressings
whereby a mechanically stable mesh can be applied in direct contact with the wound while a
hydrocolloid device is positioned above it to absorb exudates [5].

However, despite the global dimension of the clinical problem, there has been limited progress in developing wound dressings with reliable healing properties and still a large number of patients suffer of negative clinical outcomes. In particular, it is argued that the development of new devices has not adequately taken into account the wide physicochemical, biochemical and cellular differences occurring in the wound bed. When compared to acute wounds, chronic wounds are affected by a persistent inflammatory environment leading to increased exudate volumes, increased wound and peri-wound temperature, to a more alkaline wound environment (7.15 to 8.9) and to different cellular behaviours [12]. Furthermore, although biomaterials widely considered safe are used for the manufacturing of wound dressings, the host response that any implant triggers when in contact with body fluids has thus far been neglected despite in depth studies have been advocated [13]. Indeed, the early adsorption of proteins of the exudate can affect the adhesion and activation of immune cells as well as the behaviour of tissue cells and potentially alter the tissue regeneration process at the biomaterial/tissue interface [14].

Consequently, the present *in vitro* work aims to provide a systematic *in vitro* investigation of the interactions occurring between seven types of clinically-available wound dressings, distinct in their biomaterial composition and engineering, with the biochemical and cellular components of the exudate and to ascertain the role that these interactions may play on the wound healing process.

In particular, the biomaterial physicochemical properties (e.g. changes in the dressing swelling properties and water retention capability) when exposed to simulated body fluid or human plasma were linked to changes in inflammatory cell adhesion and cytokine release.

## 2. Material and methods

### 2.1 Preparation of wound dressings

Seven different groups of clinically-available wound dressings were selected and divided in three categories to cover the range of biomaterial composition, engineering mostly adopted in clinics (Table 1): Group A, non-resorbable synthetic polymers, Group B: non-resorbable

**Table 1. Wound dressing categories, their composition and recommended use.**

| Group A: Non-resorbable synthetic or semi-synthetic polymers | Group B: Non-resorbable natural polymer (i.e., cellulose and carboxymethyl cellulose-based dressing) | Group C: Resorbable natural polymer (i.e., alginate-based dressing) |
|---|---|---|
| **Atrauman (Hartman AG, Germany)** Non-medicated tulle dressing consisting of a water-repellent polyester tulle impregnated with fatty acids (does not contain paraffin). *primary dressing to be used with a secondary–wound bed protection **Melolin (Smith & Nephew, UK)** Film: polyethylene terephthalate. Absorbent layer: mixture of cotton and polyacrylonitrile fibres, backed with a layer of an apertured non-woven cellulose fabric. *lightly to moderately exuding wounds **Kerramax** (Crawford Healthcare, UK) *moderately to heavily exuding wounds Inner super-absorbent pad—viscose and polyester non-woven, coated with an adhesive of vinyl acetate and ethylene copolymer, with cross-linked sodium polyacrylate particles sandwiched. Outer non-woven Layers–thermo-bonded non-woven, made of polyethylene/ polyethyleneterephthalate fibres. | **Aquacel (Convatec, US)** Sodium CMC (hydrocolloid polymer) is spun into fibres and then made into both sheet and ribbon dressings. *Moderately to heavily exuding wounds **Kerracel (Crawford Healthcare, UK)** 100% CMC gelling fiber technology that transforms into a soft gel upon contact with wound exudate. *moderately to heavily exuding wounds **N-A Ultra (Medical Dressings, UK)** Film: Silicone Absorbent layer: primary wound contact layer consisting of a knitted viscose rayon sheet. Rayon is a semi-synthetic polymer derived from the degradation and regeneration of cellulose *primary dressing to be used with a secondary–wound bed protection | **Kaltostat (Convatec, US)** Absorbent fibrous fleece composed of the sodium and calcium salts of alginic acid in the ratio of 80:20. *used to stop bleeding |

natural polymers (i.e. carboxymethyl cellulose, CMC), Group C: resorbable natural polymers (i.e. alginate). In all experiments, samples of each group were prepared by careful cutting of 1.5x1.5 cm squares.

## 2.2 Fourier-transformed Infrared spectroscopy (FTIR)

Wound dressings were analyzed by a FT-IR spectrometer (Spectrum Two™ FT-IR Spectrometer and Analysis Systems, PerkinElmer, UK) whereby spectra of each sample were obtained in the range of 300–4000 $cm^{-1}$ at a scan resolution of $4cm^{-1}$.

## 2.3 Scanning electron microscopy (SEM)

Scanning electron microscopy of the dry and swollen wound dressings was performed by variable pressure scanning electron microscopy (Environmental Scanning Electron Microscope EVO SEM, Zeiss, Germany) after 24h incubation at 37˚C in either Radin's simulated body fluid (SBF) [15] or human plasma. The analysis enabled the study of the morphological and swelling properties of the fibers in absence and presence of biological macromolecules. Swollen samples were mounted on SEM stubs without sputter coating and analysed at the same humidity and variable pressure ranging from 478 to 534Pa, at different magnifications. Fibre diameter was measured on 2 fibres from three samples for each type of dressing (n = 6) in the different conditions.

## 2.4 Swelling test

The swelling kinetics of each dressing were assessed by gravimetric analysis over 24 hours in the three different conditions: (i) dressing incubated in Radin's simulated body fluid (SBF), (ii) dressing incubated in normoglycaemic human plasma (supplier, country) and (iii) dressing incubated in hyperglycaemic human plasma obtained by dissolving glucose in the human plasma to reach a 22 mg/ml concentration. Sample weights over time was measured to provide the actual water absorption content of each dressing and give comparable indications of the kinetics of swelling across woven and non-woven devices.

The percentage of swelling was also calculated according to the formula

$$Swelling(\%) = \frac{Ws - Wd}{Wd} x \, 100$$

Where Ws is the weight of the sw sample and Wd is the weight of the dry sample.

Dressing macroscopic evaluation was also performed to show changes in dressing morphologies.

## 2.5 Moisture vapor transmission rate (MVTR)

The ability of the wound dressings to retain water was analysed by an established method of moisture vapor transmission rate (MVTR) [16]. In those cases where dressings were made of more than one layer, the experiments were also performed on samples of the absorbent layer where their film had been removed. This enabled the study of the water retention properties of the whole medical devices as well as of their absorbent layer(s). Samples were placed in plastic containers and swollen in either normoglycaemic or hyperglycaemic human plasma for 10 min to mimic the initial contact with the wound exudate. They were then drained by minimal contact with tissue papers to remove the excess of incubation medium and weighted to obtain the cumulative weight of the sample and its container. Samples (n = 3) were incubated in static conditions at 34˚C, for 24 h and finally weighted again to measure the water loss. Data were

expressed as MVTR according to the formula.

$$MVTR\left(Evaporated\ water\ \frac{volume}{T}\right) = (Ws - We)/t$$

Where Ws is the initial cumulative weight of the sample and container (mg) after the initial 10 min swelling, We is the weight (mg) of the sample and container after evaporation, and t refers to the test period (i.e. 24h).

## 2.6 Protein adsorption

The adsorption of serum proteins on the dressing fibres was analysed by sodium dodecyl sulphate poly(acrylamide) gel electrophoresis (SDS-PAGE). Briefly, wound dressing samples (1cm x 1 cm) were incubated for 1h in 0.5 mL of foetal calf serum, 37˚C, static condition. In the case of multi-layered dressings, the protective polymeric film was removed to focus the study only on the absorbent layer(s). Samples were then copiously washed in sterile deionised water to remove proteins absorbed within the dressing mesh or loosely bound to their surface. Following this step, the dressings were incubated in 50 μL of electrophoresis sample buffer (BioRad, Watford, UK) for 1h, room temperature. Electrophoresis was performed on 10 μL samples according to a standard 10% SDS-PAGE using a Mini Protean electrophoresis kit (BioRad). Molecular weight standards (PageRuler Plus, Prestained Protein Ladder, Thermoscientific, Vinhus, Lithuania) were used as a reference. The electrophoresis was performed at 100 mV for 2 h. To enhance the detection of protein species bound to the fibres in low amounts, a sensitive silver staining (BioRad, Watford, UK) of the gels was adopted. Protein electrophoretic profiles were documented by image analyser by visible light (BioRad, Watford, UK).

## 2.7 Inflammatory cell adhesion

U-937 cells (European Collection of Authenticated Cell Cultures, Catalogue no: 85011440) are a pro-monocytic model human cell line. Cells were expanded and passaged in DMEM medium (Gibco, UK) containing 10% v/v human plasma (Gibco) and maintained at 37˚C in humidified atmosphere with 5% $CO_2$ for 3 days. Wound dressing samples were cut to a 1.5×1.5cm size under sterile conditions and placed in separate wells of a 24-wells plate containing 2ml of 10% v/v human plasma-enriched DMEM or 2ml of the same medium enriched with 10% v/v human plasma in either normoglycaemic or hyperglycaemic conditions. U937 cells (100,000/dressing sample) were seeded on the absorbent layer of each dressing (n = 3). Controls were prepared where U937 cells were seeded in empty wells. All samples were kept in incubator for 24 hours, 37˚C, static conditions.

The samples were washed and their nuclei stained by DAPI staining. The differentiation of the cells into either macrophage pro-inflammatory (M1) or post-inflammatory, regenerative M2 phenotype was analysed by immunostaining. Briefly, samples were incubated in 3% w/v bovine serum albumin (BSA, Sigma Aldrich, UK) in phosphate buffered saline pH 7.4 (PBS, Sigma Aldrich, UK) for 1h at room temperature to block any non-specific binding of the antibodies used for immunostaining. After the blocking step, samples were washed three times in PBS and incubated in either a 1:100 3% w/v BSA solution of iNOS mouse anti-human antibody or of CD206 rabbit anti-human antibody (Abcam, UK) for 3 h, room temperature. A secondary FITC-conjugated goat anti-rabbit antibody (Abcam, UK) solution diluted 1:100 in the same blocking solution was added for 1h, room temperature. After washing in PBS, samples were stored in PBS at 4˚C until analysed by two epi-fluorescent confocal microscopes; the Leica TCS SP5 to yield DAPI staining images merged with bright field images and Zeiss

2633000523 to obtain high magnification of immunostaining. Microscopy analysis was performed at 20x magnification and different zoom settings.

## 2.8 Statistical analysis

All data were analysed by ANOVA t test from n = 3. In the case of ELISA data, statistical analysis was performed with a one-way ANOVA test comparing simulated normoglycaemic versus hyperglycaemic conditions for each type of wound dressing, followed by Tukey's test across different wound dressings.

## 3. Results

For the purpose of linking the dressings polymer composition and engineering to the swelling and water retention tests as well as to the inflammatory cell adhesion and activation profiles, the FTIR analysis mainly focussed on the presence of hydroxyl (-OH) groups as indicators of the biomaterial hydrophilicity. Fig 1 clearly shows that, among the wound dressings with absorbent layers made of synthetic polymers, only Kerramax showed a significant presence of -OH groups mainly ascribed to the biomaterial polyacrylic acid moieties. Among the synthetic biomaterials, Atrauman, Kerramax and Melonin showed peak profiles different from each other in the region 500 cm$^{-1}$ to 1800 cm$^{-1}$, with only the C = O peak being clearly present in all

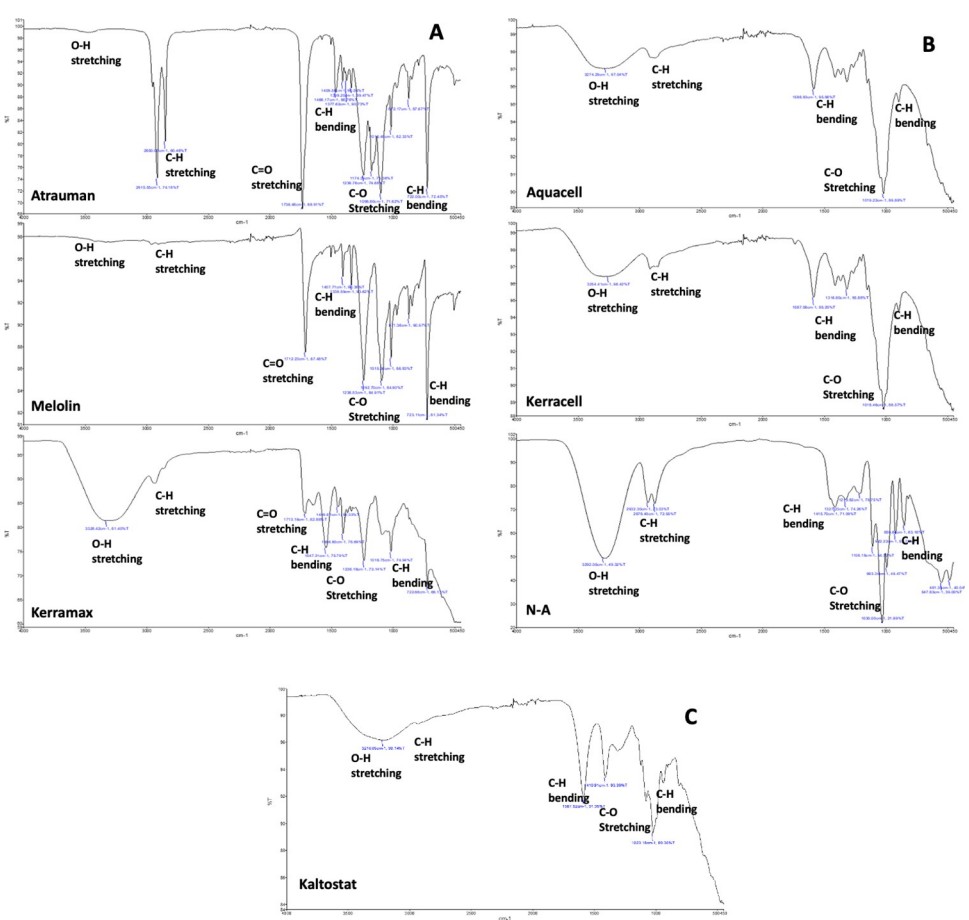

**Fig 1. FTIR of the absorbent layers of the wound dressings under examination. A.** synthetic polymers, **B.** cellulose derivatives, **C.** alginate.

of them. Atrauman also shared two peaks at 2915 cm$^{-1}$ and 2850 cm$^{-1}$ identified as -CH stretch in the synthetic polymeric chain. Among the cellulose-based dressings, large -OH bands were found in the N-A dressing due to the presence of the -CH$_2$OH groups of the Rayon that is a semi-synthetic material obtained from the regeneration of degraded cellulose (see Table 1). The remaining carboxymethyl cellulose (CMC)-based wound dressings also showed the typical -OH peak at 3,400 cm$^{-1}$ that was always accompanied by two smaller peaks at 2,900 cm$^{-1}$. As expected, the regenerated cellulose-based N-A as well as the CMC-based wound dressing showed similar profiles in the region 500 cm$^{-1}$ to 1800 cm$^{-1}$. Likewise, the alginate-based Kaltostat showed only the large -OH peak at 3400 cm$^{-1}$ and typical bands of polysaccharides in the region 500 cm$^{-1}$ to 1800 cm$^{-1}$.

SEM analysis of the swollen wound dressings was performed by variable pressure SEM analysis of the dry and swollen fibres in either Radin's SBF or human plasma. Fig 2 shows that in the case of the relatively hydrophobic Melolin, the fibres tended to coalesce, but their diameter did not significantly change. A similar behaviour, but less marked than Melonin, was observed in Atrauman where the fibre diameter slightly increased as expected from the presence of a relatively small -OH band observed by FTIR that suggested their relative limited hydrophilicity. Consistently to the FTIR results indicating the relatively high presence of -OH groups, Kerramax fibres significantly increased their diameter and tendency to coalesce in SBF. In this relatively hydrophilic biomaterial, the analysis of the fibre behaviour in human plasma was made difficult by the overall swelling behaviour of the dressing that enabled the observation of only few fibres emerging from the liquid bulk. Similar observations were made in the case of the CMC- and alginate-based dressings. Aquacell, Keracell and Kaltostat showed a significant increase of the diameter of the fibres as well as their tendency to coalesce in SBF and to retain large volumes of human plasma making their assessment in these conditions difficult. A different behaviour was observed in N-A dressings where the regenerated and knitted cellulose fibres, that were tightly aligned in dry conditions, tended to undergo only a relatively limited swelling and to become separated in SBF. Overall, while the swelling in SBF and HP did not change significantly in the hydrophobic dressings, the diameters of the fibres in hydrophilic biomaterials showed tendency to a reduced swelling in HP (Fig 2D).

The results of the swelling rate with SBF (data not shown) and HP showed no significant difference in any of the wound dressings examined. However, differences were observed when swelling in simulated normoglycaemic and hyperglycaemic human plasma was monitored (Fig 3). Kaltostat, showed the highest and fastest swelling that amounted to 80% of its initial weight in normoglycaemic HP with the plateau reached after 7h and up to 100% in just 1 h in hyperglycaemic HP. Kerracel and Kerramax showed the second highest swelling properties swelling up to ca 70% in 24h in both normoglycaemic and hyperglycaemic conditions. These data reflected the presence of large -OH FTIR bands and fibre swelling in Kerracel, but not in Kerramax where, despite the presence of -OH groups, the fibres did not show any significant swelling. It can be argued that the engineering of the materials in the form of a disordered and dense CMC fibre mesh (Fig 2, Kerracel dry) or relatively more rigid woven synthetic polymer fibres (Fig 2, Kerramax dry) is an important parameter in defining the swelling properties of wound dressings. Such an effect of the device engineering is also remarked by the different behaviour of another CMC-based wound dressing, Aquacel, where the hydrophilic character of the biomaterials is combined with a relatively loser mesh of disordered fibres (Fig 2, Aquacel dry). In such a case, the overall biomaterials hydrophilic properties led to a 40% swelling in normoglycaemic conditions in 24 h, whereby the osmosis generated by higher concentrations of glucose appeared to exert a pressure on the relatively loser mesh significantly increasing the swelling to the same level (70%) of Kerracel and Kerramax (Fig 3). The role played by both the biomaterial hydrophilicity and its engineering became even more obvious when N-A, was

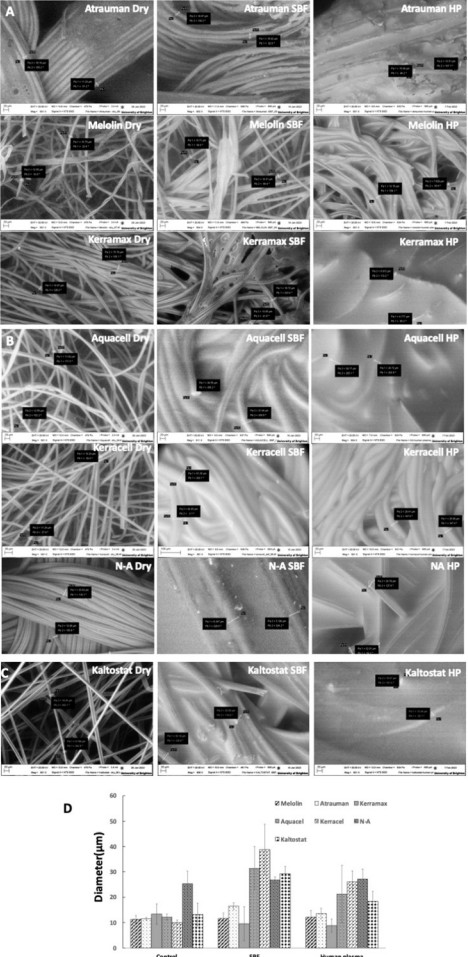

**Fig 2. Variable pressure SEM of wound dressings fibres in dry and swollen status in SBF and HP.** Typical fibres diameters under the different conditions are reported in each micrograph. Micrographs magnification was x500 for all with the exception of NA in SBF where magnification was 3.6K. Variable pressure ranging from 478 to 534Pa. **A.** Wound dressings based on synthetic polymers, **B.** Wound dressings based on cellulose derivatives, **C.** A wound dressing based on alginate, **D.** Measurements of fibre diameters in the dry, SBF-swollen and HP-swollen conditions (n = 6). Images were obtained by the Environmental Scanning Electron Microscope EVO SEM, Zeiss.

investigated. This regenerated cellulose-based wound dressing is engineered by the manufacturer as relatively tight-knitted fibres (Fig 2C, N-A dry). In the case of this type of wound dressing, the relatively hydrophilic fibres are packed and their engineering in tight aligned bundles allows only for a non-significant swelling (Fig 2C, N-A dry, SBF, HP and Fig 2D). This resulted into a relatively low level of overall swelling (30% in normoglycaemic HP and 35% in hyperglycaemic HP) slowly reaching its peak at 24 h (Fig 3). The macroscopic evaluation of all the dressings when in dry and swollen state shows how knitted mesh can maintain their morphology upon swelling, while others acquire gel properties (S1 Fig).

The effect played by the biomaterial properties and its engineering became even more evident when hydrophobic synthetic polymers such as those used for the manufacturing of Melolin and Atrauman were studied. Both these wound dressings showed a very limited swelling as a consequence of: (i) the relatively hydrophobic properties of their fibres (Fig 1, Melolin and Atrauman), (ii) no significant fibre swelling (Fig 2D) and (iii) a relatively higher swelling

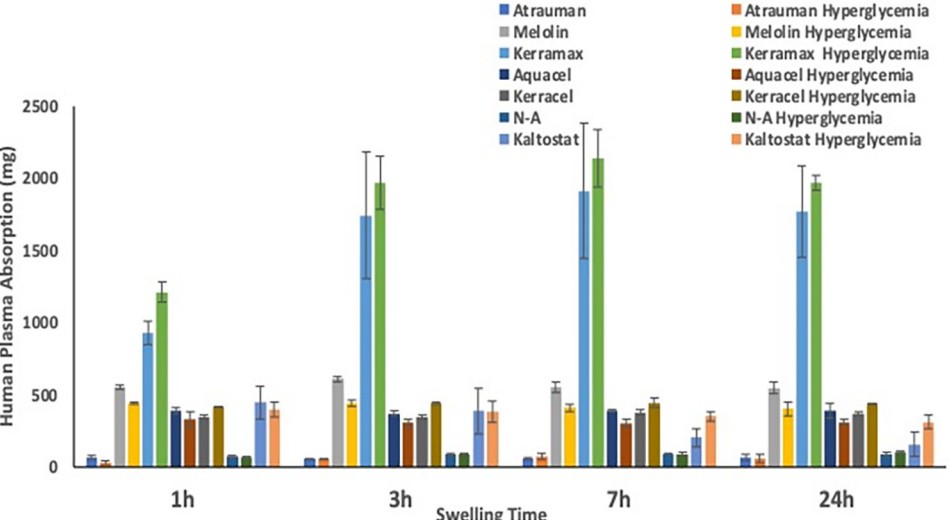

**Fig 3. Swelling behaviour of wound dressings in human plasma in simulated normoglycaemic and hyperglycaemic conditions over 24h incubation.**

caused by the glucose-driven osmotic pressure in the case of a lose mesh (Fig 2, Melolin dry and Fig 3, Melolin normoglycaemic: 9%, Melolin hyperglycaemic: 18%) when compared to tightly knitted fibres bearing some -OH groups (Fig 2, Atrauman dry and Fig 3, Atrauman normoglycaemic: 22% and Atrauman hyperglycaemic: 25%). It has also to be considered that the Melolin synthetic absorbent layer is backed up by a cellulose mesh that is likely to contribute to the absorption of exudate.

Likewise, the ability of the wound dressings to maintain moisture was clearly affected by their engineering during the manufacturing process (Fig 4). When tested in its manufactured 3-layers, Melolin showed very low evaporation despite the hydrophobic character of the absorbent synthetic polymer; this was ascribed to the presence of the silicone film and of the interposed layer of cellulose. When the test was repeated only on the Melolin absorbent layer, a significant increase of evaporation occurred unless the presence of high concentrations of glucose in the HP would result in sufficient retention possibly through the relatively higher viscosity of the HP and to the hydrogen bonding forming between the water and glucose molecules. A similar behaviour, but with a significantly higher levels of evaporation, was observed in the case of Kerramax where the relatively hydrophilic synthetic polymer was engineering into different layers where a super-absorbent and viscose non-woven polyester pad are coated with an adhesive of vinyl acetate and ethylene copolymer and cross-linked sodium polyacrylate particles and an outer film of polyethylene/ poly(ethylene terephthalate).

The knitted fibres of both Atrauman and N-A, showing relatively low levels of swelling, were subjected to low liquid loss in both simulated normoglycaemic and hyperglycamic conditions, despite their different chemical composition.

The remaining CMC-based wound dressings showed similar behaviour, with the ability of preserving the wound moist being enhanced in hyperglycaemic conditions. Kaltostat showed a similar, but statistically not significant trend.

The SDS-PAGE profiles of proteins adsorbed on the fibre surfaces of the various dressings under investigation showed differences mainly between the two tulle-based dressings and the hydrogel-based ones (Fig 5). In all cases, the most represented protein was albumin, putatively identified by its typical molecular weight of ca 66kDa (Fig 5, black arrows). While this protein

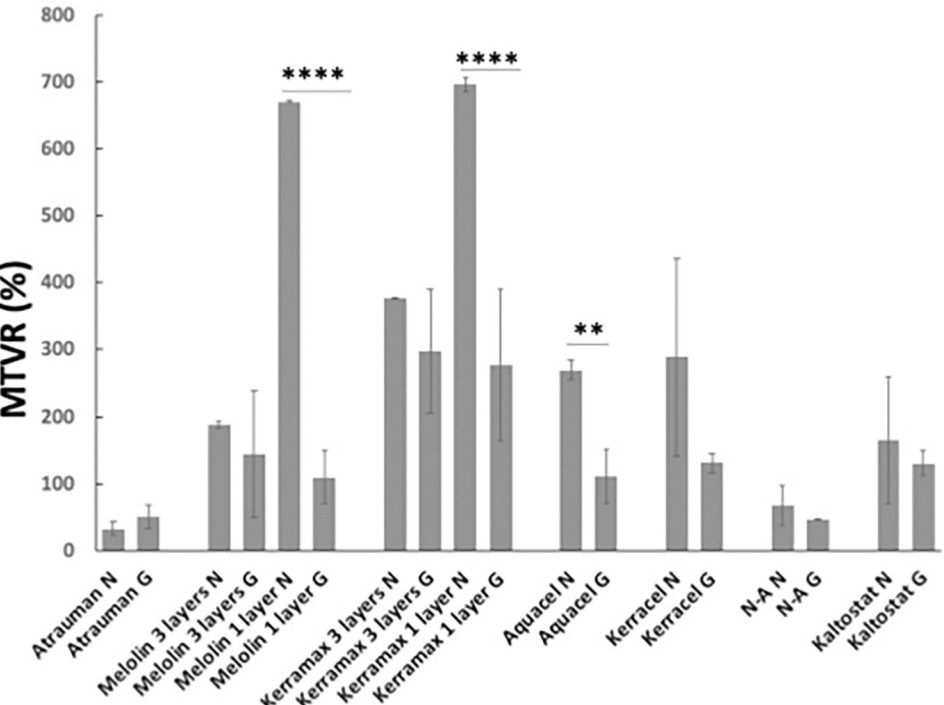

**Fig 4. Rate of evaporation from different wound dressings over 24h.** Melolin and Kerramax 3 layers show the evaporation rate in the complete dressings. Melolin and Kerramax 1 layer show the evaporation rate of the absorbent layer. N: simulated normoglycaemic conditions, G: simulated hyperglycaemic conditions. MTVR % was calculated subtracting the weight of the samples at 24h from that of the dressings swollen for 10min as measured at the start of the experiment.

was the only one adsorbing on the fibres of Atrauman (lane 2), N-A (lane 3) and Kaltostat (lane 8), the elution of proteins with higher (Fig 5, bracket) and of one main band at 20kDa (Fig 5, white arrow) was observed in the case of all the other dressings. Traces of other proteins could be observed throughout the molecular weight range.

The interactions of monocytes/macrophages with the wound dressings under investigation showed a pattern that was consistent with their physicochemical properties and, at some extent, with the protein adsorption profile (Fig 6).

Firstly, different patterns of cells interactions were observed when biomaterials identified by FTIR analysis as either hydrophilic or hydrophobic were compared. In the case of the most hydrophobic Melolin, also showing the most prominent adsorption of proteins over a wide range of molecular weights (Fig 5, lane 4), fibres were colonized by the inflammatory cells with the highest level of colonization being observed in the case of experiments performed in hyperglycaemic conditions (Fig 6, Melolin, normoglycaemia and hyperglycaemia). The other synthetic non-woven dressing, Kerramax also showed an alignment of adhering cells on the surface of its fibres, but at a lower density (Fig 5, Kerramax, normoglycaemia and hyperglycaemia). The CMC-based hydrogels (Aquacel and Kerracel), identified as highly hydrophilic by FTIR analysis (Fig 1, -OH bands) and demonstrated to bind a wide range of protein species, showed relatively lower levels of adhesion of the cells which were mainly entrapped in the porosity of these hydrogels rather than adhering on their fibres. This was particularly the case of Kerracel, the relatively higher level of fibre swelling observed by SEM (Fig 2B and 2D) significantly reduced the hydrogel porosity thus limiting cell penetration within the absorbent

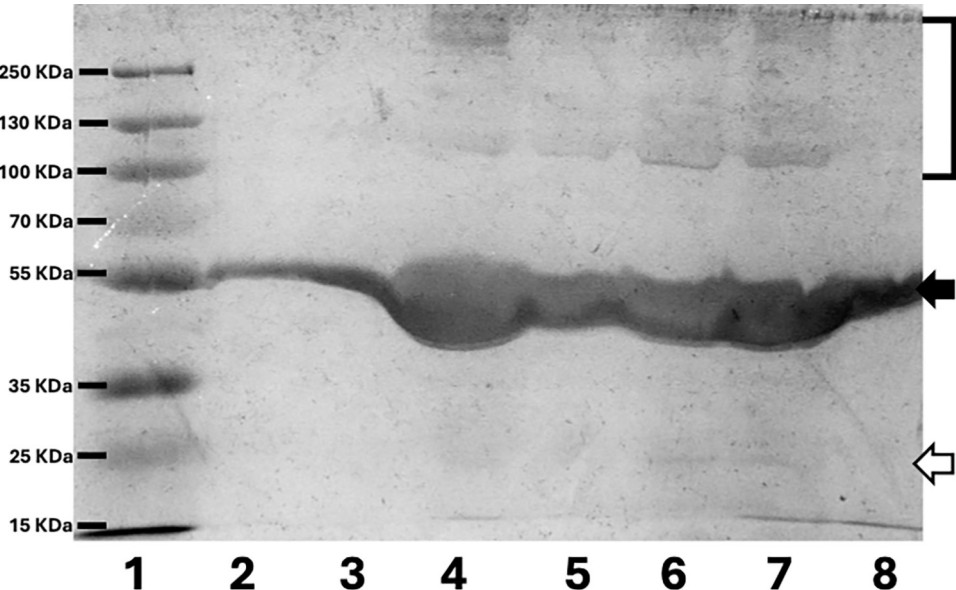

**Fig 5. Profiles of protein adsorption on wound dressings.** Image shows the electrophoresis profile of the proteins adsorbed on the surface of the studied wound dressings. Proteins were allowed to adsorb for 1h, 37°C, static conditions. After copious washing with de-ionised water, adsorbed proteins were eluted by 1h incubation with SDS-PAGE sample buffer (Biorad, UK), room temperature, static conditions. The elution was driven by the SDS disrupting physical bonding between the proteins and the biomaterial surfaces. Samples of 10 µl were uploaded onto the SDS-PAGE. Lane 1: Molecular Weight Markers, lane 2: Atrauman, lane 3: N-A, lane 4: Melolin, lane 5: Aquacel, lane 6: Kerracel, lane 7: Kerramax, lane 8: Kaltostat.

layer into small pockets (Fig 6, Kerracel, normoglycaemia and hyperglycaemia). Kaltostat, an alginate-based and highly hydrophilic dressing, showed adhesion of cells on its fibres rather than entrapment in its porosity in both normoglycaemic and hyperglycaemic conditions

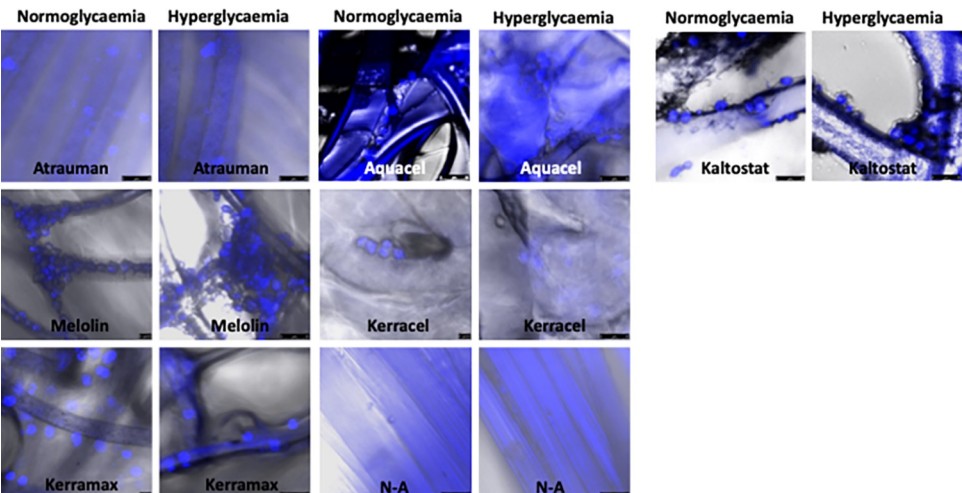

**Fig 6. Confocal microscopy analysis of the U-937 monocyte/macrophages cell line interactions with wound dressings in simulated normoglycaemic and hyperglycaemic conditions.** Wound dressing samples (1.5x1.5cm) were incubated with $1 \times 10^6$ cells/2ml for 24h in 10% (v/v) FBS-enriched RPMI medium and 10% (v/v) FBS-enriched RPMI medium with a 22mM glucose concentration. Merged images of bright field with DAPI staining allowed the analysis of the cell/biomaterial interactions distinguishing between the cells adhering on the fibre surface from those entrapped in the dressing porosities. Images taken by a Leica TCS SP5 at 20x maginfication.

suggesting that the polymer chemical composition can override the effect of the hydrophilicity and the prevalence of the passivating protein albumin in determining the biomaterial cell substrate properties.

Secondly, similar reduced cell interactions were observed in devices made of knitted fibres, irrespective of their polymeric composition. Atrauman (a dressing made of synthetic polymers) and N-A (a cellulose-based dressing) showed levels of cell interactions lower than those observed in hydrogels; this being the case in both simulated normoglycaemic and hyperglycaemic conditions. It is worth highlighting that these findings could be a consequence of the technical limitation caused by the large porosity of the knitted dressings that, upon cell seeding, would lead most of the cells to be deposited on the underlying tissue culture plastic rather than on the fibres (Fig 6, Atrauman and N-A, normoglycaemia and hyperglycaemia).

Fig 7 shows the differentiation of cells as either M1 (iNOS$^+$) pro-inflammatory phenotype or as M2 (CD206$^+$) regenerative phenotype in simulated normoglycaemic and hyperglycaemic conditions on the different wound dressings.

As already shown in Fig 6, the two knitted dressings, Atrauman and N-A showed a neglegible number of adhering cells, mostly in a M1 phenotype (Fig 7A, Atrauman and Fig 7B, N-A).

The two other wound dressings with absorbent layers made of synthetic polymers showed to induce different phenotypes in the U937 adhering cells. In particular, the hydrophobic Melolin fibres showed to support the adhesion of M1 (iNOS$^+$) macrophages in simulated normoglycaemic conditions with only sporadic presence of M2 (CD206$^+$) cells (Fig 7A, Melolin). The expressions of both the iNOS and CD206 markers were lower in the case of the hydrophilic synthetic Kerramax (Fig 7A, Kerramax).

The two CMC-based dressings, Aquacel and Kerracel showed a predominance of adhering iNOS$^+$ cells particularly in simulated hyperglycaemic conditions (Fig 7B, Aquacel and Kerracel). However, they differed in their ability to support the post-inflammatory phenotype as the latter appeared to be more present as entrapped non-adherent cells in Kerracel in both simulated normoglycaemic and hyperglycaemic conditions. In the case of Kaltostat, the degradation of this biomaterials made difficult to distinguish adhering cells from entrapped ones with the immunostaining appearing to show a prevalence of M2 macrophages (Fig 7B, Kaltostat).

U937 response to the different wound dressing biomaterials in simulated normoglycaemic and hyperglycaemic conditions was also evaluated in terms of biochemical signalling playing a role in inflammation and tissue healing (Fig 8A–8C).

All the non-woven dressings showed relatively high levels of TNFα release with no significant differences being observed between the simulated normoglycaemic and hyperglycaemic conditions, but with the exception of Kerracel where the cells showed a significantly lower release of this pro-inflammatory cytokine in simulated normoglycaemic conditions and levels similar to those of the other dressings in simulated hyperglycaemic conditions (Fig 8A). A similar pattern was observed when the released levels of TGFβ were evaluated (Fig 8B). Kerracel was the only one to differ from the others when this growth factor was measured (Fig 8B). In this case, the secretion was lower than all the other non-woven dressings, but increased in conditions of simulated hyperglycaemia, albeit in a non-significant manner. Different patterns of secretion were observed for PDGF-BB (Fig 8C). This potent stimulator of fibroblasts proliferation was released at significantly lower level by Kerracel, particularly in simulated hyperglycaemic conditions. Among the other non-woven dressings analysed, only Melolin showed levels of secreted PDGF-BB lower in simulated hyperglycaemic conditions (Fig 8C). The alginate-based Kaltostat also showed relatively low levels of all the cytokines analysed and no signficant difference between simulated normoglycaemic and hyperglycaemic conditions (Fig 8A–8C).

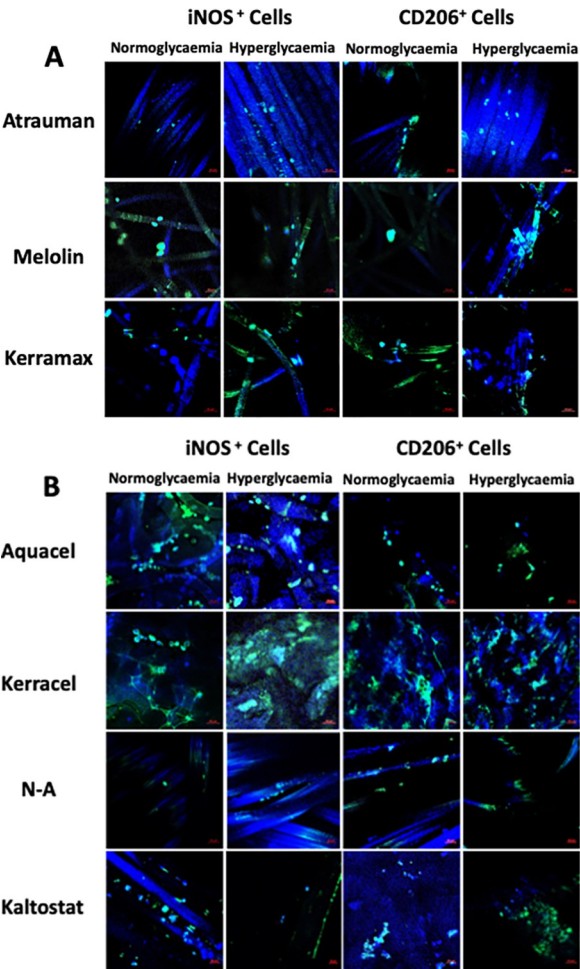

**Fig 7. Confocal microscopy analysis of U937 cell differentiation upon contact with wound dressing in simulated normoglycaemic and hyperglycaemic conditions.** Wound dressing samples (1.5x1.5cm) were incubated with $1\times10^6$ cells/2ml for 24h in 10% (v/v) FBS-enriched RPMI medium and 10% (v/v) FBS-enriched RPMI medium with a 22mM glucose concentration. **iNOS:** marker of M1 pro-inflammatory phenotype. **CD206:** marker of M2 regenerative phenotype. Micrographs show merged epifluorescence microscopy images for DAPI staining (cell nuclei, blue staining) and FITC-labelled antibodies (green staining). Images were taken by a Zeiss 2633000523 confocal microscope, x20 magnification.

The results obtained when cells were incubated with the two woven dressings, Atrauman and N-A showed lower levels of all cytokines and growth factors analysed in the former (Fig 8A–8C) with a a significant decrease of TNFα release in the case of N-A (Fig 8A) and a significant increase of PDGF-BB secretion in the case of Atrauman (Fig 8C) in simulated hyperglycaemic conditions.

## 4. Discussion

When compared to other medical devices, studies linking the physicochemical properties of wound dressings to their biocompatibility and clinical efficacy lack of the widely advocated systematicity [3, 13, 17]. Rather, most of the published studies focus on the devices healing properties based either on clinical evaluation parameters or on pre-clinical histological assessment [18–22], while their physicochemical properties have been investigated in conditions not reflective of the wound bed environment [23, 24]. To study the biocompatibility of wound

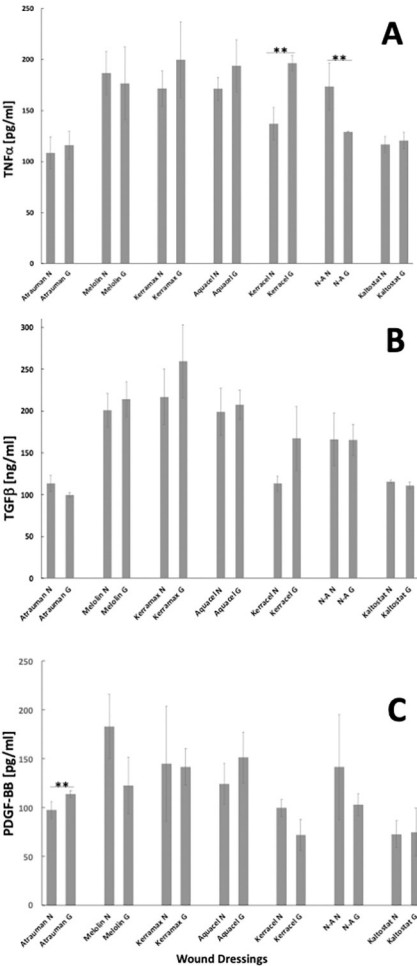

**Fig 8. Levels of TNFα, TGFβ and PDGF-BB released by U937 monocytes/macrophages upon contact with wound dressings over 24h incubation. A**: TNFa, **B:** TGF-β, **C:** PDGF-BB. Experiments were performed by seeding $10^6$/cells/ml/1.5x1.5 cm dressing coupons for 24h in either simulated normoglycaemic or hyperglycaemic medium.

dressing biomaterials, this work has focussed on wound dressings widely used in clinics that have no excipients (e.g. antibacterial agents) which could have altered the interactions of proteins and immunocompetent cells with the dressing biomaterial. When non-woven dressings such as Aquacel (a dressing with a carboxymethyl cellulose absorbent layer), Melolin (a dressing with a polyacrylonitrile absorbent layer backed by a tight cellulose layer) and Kaltostat (an alginate-based dressing) were analysed for their swelling properties in simulated body fluids mimicking normoglycaemic and hyperglycaemic conditions, they showed values different from those observed in literature. In particular, Kaltostat and Aquacel reached a plateau of over 80% and 50% wt/wt after 24 h in normoglycaemic conditions and of 100% and 70% wt/wt in hyperglycaemic conditions as compared to approximately 20% (Kaltostat) and 19% wt/wt (Aquacel) observed by Uzun et al. [23] and of 25% wt/wt (Kaltostat) and 15% wt/wt (Aquacel) observed by Minsart et al. [24]. This higher swelling properties were also matched by less vapour loss after 24h when compared to the results of Minsart et al. [24]. An opposite trend was observed when the percentage of Melolin swelling (less than 10% wt/wt in normoglycemic conditions, over 24 h) was compared to that observed by Uzun et al that was 18% wt/wt [23].

The measurement of the fibre diameters in dry conditions for these three dressings were in ranges comparable to those found by Uzun et al. [23] and slightly differed in SBF-swollen conditions arguably because of the different setting parameters applied for the environmental SEM analysis. This suggested that the differing swelling data may be due to different experimental set ups. However, trends of swelling were the same showing Kaltostat as the biomaterial able to absorb and retain more exudate [9, 21]. When analysed in relation to their chemical composition (i.e. FTIR analysis), engineering (i.e. woven fibres or hydrogels) and fibres' swelling data (i.e. variable pressure SEM), it appears that the overall swelling of the wound dressings is more affected by their engineering than by the biomaterial chemistry. Indeed, not all the dressings made of relatively hydrophilic biomaterials, as determined by the presence of -OH stretch peaks in the FTIR spectra, show high levels of swelling. In particular, the overall swelling of the relatively hydrophilic CMC-based hydrogels Aquacel, Kerracel was not significantly higher than the levels found for relatively hydrophobic biomaterials like Melolin. This could be explained both by the presence of hydrophobic domains generated by the carboxymethyl residues and by the presence of an interposed layer of cellulose in the case of Melolin. Likewise, the two fabric-engineered dressings, the relatively hydrophobic Atrauman and the hydrophilic cellulose-based N-A show equally low levels of overall swelling in human plasma even if, as expected, the fibres in N-A swell more than those of Atrauman. Consistently with the interpretation that engineering design is the most relevant parameter in determining the dressings' swelling properties, it is note that, although showing relatively high biomaterial hydrophilicity and the highest level of overall swelling, the fibres of Kerramax do not significantly increase in diameter when immersed either in SBF or human plasma rather indicating a high capacity of its fibres to disentangle upon liquid uptake.

The fibre engineering also appeared to be the main factor determining the adsorption of specific protein species as well as the adhesion of inflammatory cells. Although Atrauman and N-A are made of different types of biomaterials, their engineering into knitted fibres led to a protein adsorption limited to albumin, a protein known to 'passivate' the surface of biomaterials making them less susceptible to cell adhesion. Other proteins species, particularly in the high molecular weight range, were observed to be more represented in the CMC-based hydrogels Aquacel, Kerracel and Melolin; the latter having an intermediate layer made of CMC, too. As stated above, the carboxymethyl residues may represent hydrophobic domains where different protein species can establish hydrophobic interactions, unlike the N-A regenerated cellulose (also known as rayon) that is more hydrophilic. Among the hydrogel-based biomaterials, Kaltostat was the only one able to minimise protein adsorption to albumin as well as Atrauman and N-A. Taken together, the data of the present study support the findings of a previous work linking the adsorption of protein on different surfaces [25]. It has been shown that albumin adsorption is enhanced on hydrophilic carboxyl surfaces while high molecular weight proteins such as IgG were observed on hydrophobic surfaces. However, data of macrophage adhesion and activation are in disagreement with previous findings showing that hydrophilic surfaces enhance anti-inflammatory factors while hydrophobic surfaces promote the production of inflammatory cytokines [25]. It is widely accepted that the progression of the inflammatory response plays a key role in determining the healing of the wounds or its chronic status [1]. In particular, the role of macrophages and their change of phenotype from a pro-inflammatory (M1) phenotype to a regenerative (M2) phenotype seems to be a crucial step for the healing process [26–29]. Therefore, it is reasonable to argue that the contact of the wound dressing absorbent layers with the wound bed can create either an environment favourable to this transition or to trigger a foreign body response leading to a chronic inflammation [30]. In fact, even if dressings are periodically changed (typically every 3 days in non-diabetic wounds and 7 in diabetic cases), their contact with the wound bed can be protracted for over

two weeks thus becoming able to affect the activity of the inflammatory cells [30]. In addition, the differences observed in the physicochemical properties of the polymers used by the manufacturers and the device swelling profiles in simulated normoglycaemic and hyperglycaemic conditions appear to play a role in the ability of the dressings to capture these inflammatory cells from the wound bed and determining phenotypic changes and activation of biochemical pathways relevant to the healing process [31, 32]. It can be speculated that devices usually applied to the wound as secondary dressings because of their ability to absorb higher exudate volumes can also contribute to remove inflammatory cells from the wound bed, a process that could be particularly efficacious in removing M1 macrophages from wounds, like those of diabetic patients, prone to become chronic [33]. It may also be reasonably argued that this process could be favoured by dressings with high levels of swelling. However, although in this study no quantitative assessment was carried out, it appears that the higher levels of swelling observed, particularly in simulated hyperglycaemic conditions, do not necessarily lead to a higher penetration of cells in all the devices investigated. This seems indeed to be the case for the CMC-based wound dressings, Aquacel and Kerracel; in the latter the high level of fibres' entanglement obstructs the hydrogel porosity thus limiting the cell penetration to some clusters found in pockets between fibres (Figs 2 and 6) [34]. The alginate-based wound dressing, Kaltostat, uniquely combines the highest level of swelling with the ability to entrap relatively higher numbers of cells [21]. Together with the well documented fast degradation of this type of wound dressing, Kaltostat could emerge as a device able to create an environment favourable to healing in the wound bed, particularly in the case of diabetic subjects where any disturbance to the healing process is minimised by reducing the change of dressings. In other words, the degrading fibres of Kaltostat could remain deposited in the wound bed supporting tissue regeneration [21].

In this work, iNOS and CD206 were chosen among the available markers of M1 and M2 phenotype because of the particular focus on the evaluation of the release of aggressive species such as the nitric oxide free radicals (iNOS$^+$ cells) or for the role played in healing (CD206$^+$ cells). In particular, elevated levels of free radicals, as those expressed in hyperglycaemic conditions [24], have been reported to hindrance wound healing and participate in the synthesis of collagen through TGFβ [26, 27]. However, Kerracel showed to induce relatively high expression of iNOS in cells, but the levels of released TGFβ were relatively lower than other dressings and increased only at higher glucose concentrations (Figs 6B and 7B). CD206 was chosen as a favourite marker of M2 macrophages as it has been reported that these types of cells stimulate fibroblast proliferation through TGFβ and transdifferentiate into fibrocyte-like cells able to synthesise collagen [28, 29]. As reported in literature, the transition from M1 to M2 process is relatively slow and gradual thus explaining why both iNOS$^+$ and CD206$^+$ cells could be found on most of the dressings [30]. The likelihood of different macrophage phenotype subsets is even more probable in the case of wound dressings where cells can be at the same time suspended in the aqueous medium of the dressing pores or adhering to their fibres (Fig 6). Although not at the same extent and with a clear link to the fibres swelling properties, the results seem to show a prevalence of entrapped, non-adhering CD206$^+$ cells particularly in Kerracel.

To obtain a more comprehensive understanding of the effect of the biomaterials on U937 cell activation pathways, the study of TNFα, TGF-β and PDFG-BB release were studied in the cell culture supernatants. Here, it appears that the levels of all the factors analysed were more elevated in the non-woven biomaterials where the effect of the biomaterials seem to be predominant over the increase of glucose levels. The only exception was Kerracel where, alongside higher levels of iNOS expression, high glucose levels seemed to induce significantly higher levels of pro-inflammatory signalling involved in the acute (TNFα) and chronic (TGFβ) inflammation and lower levels of the fibroblast-inducer PDGF-BB. In this case, the relatively higher glucose levels led to the release of these growth factors at levels similar to those of all the other

non-woven wound dressings. The data obtained from Atrauman, N-A and Kaltostat were difficult to interpreter because of the low adhesion levels of cells in the woven dressings and the degradation of the alginate-based biomaterial where cell activation could be affected by the tissue plastic in which the experiment was performed. However, it appears that when fibres with the same woven features, but different biomaterial composition were compared, the more hydrophilic cellulose-based N-A led to a higher release of the pro-inflammatory stimuli, but with TNFα release being significantly reduced in simulated hyperglycaemic conditions.

The data of the present in vitro study are also corroborated by our recent clinical study showing similarly low levels of pro-inflammatory M1 macrophage adhesion and higher M2 phenotypes on Atrauman fibres, whereas relatively higher levels of M1 cells were observed in the case of Melolin [35]. However, this clinical study also showed that this was dependent on the patients underlying wound conditions.

## 5. Conclusions

As wound dressings are chosen by the clinicians on the basis of their experience and guidelines based on wound visual features, the analysis of the physicochemical properties and biocompatibility of commercially-available wound dressings may determine different criteria of choice and pave the way towards the development of new dressings. The results of this work appear to show that woven dressings, mainly used as primary dressings for wound protection, do not change their swelling properties in the case of higher levels of glucose and they have a limited interaction with inflammatory cells regardless of their synthetic or natural composition. When analysed in the light of recent clinical study [35], it appears that Atrauman is preferrable to N-A in terms of reducing the differentiation of monocytes/macrophages into a pro-inflammatory M1 phenotype. In the case of synthetic dressings with a non-woven absorbent layer like Melolin and Kerramax, their chemistry and engineering seems to maintain relatively open their mesh and enable the adhesion of inflammatory cells with relatively high levels of pro-inflammatory cytokines being released rather than free radical species. On the contrary, among the CMC-based dressings Kerracel seems to offer properties more suitable to a broader range of clinical applications as its absorbs relatively higher levels of exudate particularly in the presence of high glucose levels and limits evaporation while minimising the interactions with inflammatory cells that are mainly activated to produce free radicals necessary to reduce infection rather than to release factors involved in the development of chronic wounds. Likewise, the alginate-based Kaltostat combines a higher levels of swelling with the uptake and mild activation of the inflammatory cells and fast degradation; these are features allowing frequent dressings changes to remove excess of both exudate and inflammatory cells, even in the case of the fragile tissue of diabetic patients.

Therefore, the present work advocates the development of primary, tulle-based dressings made of biomaterials that, while protecting the wound from mechanical insults, would be able to induce the polarization of M2 macrophages at the interface with the wound bed. At the same time, secondary dressings with occlusive and M1 inhibition properties should enable the removal of the excess of exudate while preventing the chronicization of the inflammatory process. The combined application of these two types of dressings to a wound, particularly in hyperglycaemic conditions, would add a control of healing at cellular level.

## Supporting information

**S1 Fig. Macroscopic evaluation of knitted and hydrogel wound dressings.** A: Dressings made of synthetic polymers, B: Carboxymethylcellulose- and cellulose-based dressings; C:

Alginate-based dressing.
(TIF)

## Author Contributions

**Conceptualization:** Rachel Forss, Lucy Redhead, Matteo Santin.

**Data curation:** Shirin Saberianpour, Gianluca Melotto, Rachel Forss, Susan Sandeman, Nadia Terrazzini, Dipak Sarker, Matteo Santin.

**Investigation:** Shirin Saberianpour, Gianluca Melotto, Rachel Forss.

**Methodology:** Shirin Saberianpour, Gianluca Melotto, Nadia Terrazzini, Matteo Santin.

**Writing – original draft:** Matteo Santin.

**Writing – review & editing:** Lucy Redhead, Susan Sandeman, Nadia Terrazzini, Dipak Sarker.

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
