## [Decision Letter · Decision Letter 0]

5 Nov 2024

PONE-D-24-27036

A Systematic In Vitro Study of the Effect of Normoglycaemic and Hyperglycaemic Conditions on the Biochemical and Cellular Interactions of Clinically-available Wound Dressings with Different Physicochemical Properties

PLOS ONE

Dear Dr. Santin,

Thank you for submitting your manuscript to PLOS ONE. After careful consideration, we feel that it has merit but does not fully meet PLOS ONE’s publication criteria as it currently stands. Therefore, we invite you to submit a revised version of the manuscript that addresses the points raised during the review process.

Dear Authors,

Thank you for submitting your manuscript (PONE-D-24-27036) titled "A Systematic In Vitro Study of the Effect of Normoglycaemic and Hyperglycaemic Conditions on the Biochemical and Cellular Interactions of Clinically-available Wound Dressings with Different Physicochemical Properties" to PLOSONE. We appreciate your efforts in investigating the effects of various wound dressings under different glycemic conditions and recognize the potential relevance of your findings for wound care.

Based on the reviewers’ comments and editor’s recommendations, we believe that the manuscript requires substantial revisions before it can be considered for publication in PLOSONE. You are requested to carefully address the points raised by the reviewers to improve the scientific depth and clinical relevance of your study.

We encourage you to enhance the scientific rigor of your study by incorporating these additional analyses, clarifying your methodology, and providing a more robust discussion on the clinical relevance of your findings. Please ensure that all sections of your revised manuscript are clear, concise, and fully address each reviewer comment.

Please find the comments from both the Reviewers mentioned at the end of this email.

We look forward to receiving your revised manuscript.

Kind regards,

Amruta Naik

Academic Editor

PLOS ONE

Journal Requirements:

3. Thank you for stating the following financial disclosure: “UKRI Engineering and Research Council grant n. EP/W023164/1”.

4. Please note that your Data Availability Statement is currently missing the repository name and/or the DOI/accession number of each dataset OR a direct link to access each database. If your manuscript is accepted for publication, you will be asked to provide these details on a very short timeline. We therefore suggest that you provide this information now, though we will not hold up the peer review process if you are unable.

7. Please amend your manuscript to include your abstract after the title page.

Reviewers' comments:

Reviewer's Responses to Questions

**Comments to the Author**

1. Is the manuscript technically sound, and do the data support the conclusions?

Reviewer #1: Partly

Reviewer #2: Yes

2. Has the statistical analysis been performed appropriately and rigorously? 

Reviewer #1: No

Reviewer #2: No

3. Have the authors made all data underlying the findings in their manuscript fully available?

Reviewer #1: Yes

Reviewer #2: Yes

4. Is the manuscript presented in an intelligible fashion and written in standard English?

Reviewer #1: Yes

Reviewer #2: No

5. Review Comments to the Author

Reviewer #1: This manuscript aims to provide a systematic in vitro investigation of the interactions occurring between seven types of clinically-available wound dressings, and to explore the role of these interactions in the wound healing process.

There are some concerns.

(1) How the in vitro setting can reveal the in vivo or even clinical circumstances. The authors might give an explanation, in order to allow the readers to understanding the clinical value of these in vitro experiments.

(2) The cellular adhesion might depend on the protein absorption on different substrates. In this case, the protein absorption on different samples might also be worthy to be investigated.

(3) For Figure 6, for the cellular population distribution, the confocal microscopy might not be enough. It is also recommended to add the flow cytometry to give a quantitative analysis.

(4) In the discussion section, the in vitro results might also be worthy to be linked with clinical performances.

(5) In the conclusion section, an ideal design of wound dressing might also be worthy to be discussed.

Reviewer #2: In this article the authors compared 7 different types of wound dressing with different composition, synthetic or natural polymer. Resorbable or not. The article has a strong focus on structure and composition and its affect on the wound healing process. The article aim to provide useful information on the efficacy of different clinically available dressings and mode of selection. My major critics is that, the article is more of an industry report for testing 7 wound dressings. It donest have depth or rationale of a scientific articles. For example, the presence of infection, tissue damage, and ongoing immune responses in a chronic wounds which could significantly influence dressing performance have ignored.

Also I like to see a rationale for selecting these specific products. are these the most commonly used dressings, or do they represent a wider range of options? There are diverse range of modern dressings with antimicrobial, drug-releasing or collagen based etc

The collected data does not provide a clear understanding of how the clinical relevance of these parameters translates into better wound healing outcomes. For example, is a higher swelling ratio always beneficial in hyperglycemic wounds? the clinical implications of these findings be discussed in more depth. Otherwise it is more of a report.

hyperglycemia influences the swelling of certain dressings, especially alginate-based Kaltostat, but the study doesn’t provide a discussion of how these differences affect healing outcomes in diabetic wounds. What is the threshold for when this behaviour may becomes problematic or beneficial in wound healing process?

The study observes that certain dressings (e.g., CMC-based Kerracel) promote iNOS expression, but it does not elaborate on whether this inflammatory reaction is excessive or within normal limits. Are these immune responses favorable or detrimental in terms of promoting wound healing?

Tests were performed using 3 samples, which seems small. Were power calculations conducted to justify the sample size? specially in the context of swelling and inflammatory responses, where variability could be high.

ANOVA followed by Tukey’s tests were used to compare dressing interactions in normoglycaemic and hyperglycaemic conditions. However, were assumptions of normality and homogeneity of variance met? alternative non-parametric tests might be more appropriate if these assumptions not met.

The paper would benefit from additional discussion on how the findings may influence the selection of dressings for diabetic foot ulcers versus other types of chronic wounds (e.g., venous ulcers).

6. PLOS authors have the option to publish the peer review history of their article (what does this mean?). If published, this will include your full peer review and any attached files.

Reviewer #1: No

Reviewer #2: No

---

## [Author Response · Author response to Decision Letter 0]

22 Nov 2024

Saberianpour et al 2024 - Response to Review Comments to the Author

The authors would like to thank the reviewers for their most valuable comments and for helping us improve the quality of our manuscript. We agree with all the comments made by the reviewers and addressed them as reported in the replies below and making amendments, highlighted in yellow, to the manuscript.

Reviewer #1:

(1) How the in vitro setting can reveal the in vivo or even clinical circumstances. The authors might give an explanation, in order to allow the readers to understanding the clinical value of these in vitro experiments.

As now reported in the revised Discussion section, we have cited a clinical study recently published by us (new reference n. 35) where a similar pattern of macrophage adhesion and differentiation into either M1 or M2 phenotype on some of the dressings here studied show comparable results. We would like to bring for the reviewer’s attention that other clinical studies had already been cited in the Discussion section.

(2) The cellular adhesion might depend on the protein absorption on different substrates. In this case, the protein absorption on different samples might also be worthy to be investigated.

As requested by the reviewer, we have performed SDS-PAGE experiments to show the different profiles of protein adsorption on the surface of the wound dressings. The Materials and Methods and the Results are now reported in the revised manuscript. The SDS-PAGE results are reported as a new figure (Figure 5) of the revised paper and the data dicussed in the Discussion section in relation to the inflammatory cell adhesion and polarization.

(3) For Figure 6, for the cellular population distribution, the confocal microscopy might not be enough. It is also recommended to add the flow cytometry to give a quantitative analysis.

We agree with the reviewer that flow cytometry could provide quantitative data. However, the removal of all cells from the hydrogel mesh would not be reliable and standardised across all dressings, as it will not be possible to gurantee the complete detachment of the cells by trypsin from them because of the different swelling properties and porosity. Therefore, we respectfully ask the reviewer to accept the confocal data where we merge fluorescence microscopy images with bright fileld(not light microscopy as reported by the reviewer) images as the most appropriate method of characterisation of adhering/entrapped cells.

(4) In the discussion section, the in vitro results might also be worthy to be linked with clinical performances.

The revised discussion section now provides links to clinical data previously published by us and others. The new reference n. 35 specifically focus on the clinical data obtained by us from two of the biomaterials here studied.

(5) In the conclusion section, an ideal design of wound dressing might also be worthy to be discussed.

In the revised conclusion section, we had already provided our views about the desirable physicochemical properties of dressings to control biomaterial biocompatibility and clinical performance. We have now added a paragraph where our views about the ideal properties of primary and secondary wound dressings are emphasised.

Reviewer #2:

My major critics is that, the article is more of an industry report for testing 7 wound dressings. It does not have depth or rationale of a scientific articles. For example, the presence of infection, tissue damage, and ongoing immune responses in a chronic wounds which could significantly influence dressing performance have ignored.

We agree with the reviewer that the paper required a more in-depth rationale in the discussion section. For this purpose, the rationale of the study, that had already been reported in the Introduction section, has been reiterated in the Discussion. In addition data have been discussed more in depth by linking more clearly the physicochemical properties of the different wound dressings to the new experiments of protein adsorption (new Figure 5) and, in turn, to inflammatory cells' adhesion and differentiation. Likewise, in the discussion section, we have now added data interpretations and citations of clinical studies performed by our group in addition to the papers by other authors that had already been cited in the previous version. This link shows how the in vitro results of the present paper reflect the behavior of these wound dressings in clinics. We want to clarify that we deliberately omitted from our study dressings including antibacterial agents as we wanted to analyse specifically the effect of the materials properties on inflammatory cells behaviour; this has also stated clearly in the new version of the Discussion. Other conditions, such as those suggested by the reviewer would be difficult to simulate closely in vitro and could indeed mask the effect of the biomaterials characteristics. We respectfully bring for the reviewer’s attention that, by analysing the dressings physicochemical properties and inflammatory cell behaviour in conditions simulating both the normo- and hyper-glycaemic conditions, we have already provided indications of the performance of dressing in specific clinical conditions of the wound (i.e. diabetes).

Also I like to see a rationale for selecting these specific products. are these the most commonly used dressings, or do they represent a wider range of options? There are diverse range of modern dressings with antimicrobial, drug-releasing or collagen based etc

As stated above, we deliberately omitted any product, including additives such as drugs and antimicrobial agents. The choice of the wound dressings was made following the advice of clinicians using them in their clinical practice; They recommended the studied dressings as the most commonly used on both acute and chronic wounds. We think we made it clear in the Introduction, but we have emphasised in a new paragraph in the Discussion section.

The collected data does not provide a clear understanding of how the clinical relevance of these parameters translates into better wound healing outcomes. For example, is a higher swelling ratio always beneficial in hyperglycemic wounds? the clinical implications of these findings be discussed in more depth. Otherwise it is more of a report.

The revised discussion section now provides indications about the clinical implications of our data.

Hyperglycemia influences the swelling of certain dressings, especially alginate-based Kaltostat, but the study doesn’t provide a discussion of how these differences affect healing outcomes in diabetic wounds. What is the threshold for when this behaviour may becomes problematic or beneficial in wound healing process?

We respectfully bring for the reviewer’s attention that it would be very speculative to draw specific conclusion about how the behaviour of dressings in hyperglycemic conditions could affect the healing of diabetic wounds. However, as mentioned above, we had already added some clearer links between the interactions of macrophages with biomaterials in hyperglycaemia-simulating conditions that may suggest that a prevalence of entrapped M2 observed in the case of Kaltostat (Figure 6B, Kaltostat, now Figure 7 in the new version) could subtract these cells from the wound bed so depleting the wound from a type of cells important to healing. We tried to make it clearer in the new version of the Discussion and Conclusion sections.

The study observes that certain dressings (e.g., CMC-based Kerracel) promote iNOS expression, but it does not elaborate on whether this inflammatory reaction is excessive or within normal limits. Are these immune responses favorable or detrimental in terms of promoting wound healing?

As stated above, it would be difficult to reproduce in vitro conditions mimicking long term chronic wounds. In the new discussion section we have now made clear that iNOS expressing macrophages are activated during the acute phase or inflammation, but the presence of the wound dressings can protract their activation so contributing to the chronic evolution of the wounds. In addition, our data of TNFalpha released seem to show a link between high levels of glucose and higher inflammatory cell activation only in Kerracel and N-A, albeit with opposite trends.

Tests were performed using 3 samples, which seems small. Were power calculations conducted to justify the sample size? specially in the context of swelling and inflammatory responses, where variability could be high.

ANOVA followed by Tukey’s tests were used to compare dressing interactions in normoglycaemic and hyperglycaemic conditions. However, were assumptions of normality and homogeneity of variance met? alternative non-parametric tests might be more appropriate if these assumptions not met.

We agree with the reviewer's comments. However, we would like to bring for the reviewer's attention that the study used cell lines macrophages rather than primary cells to minimize individual variability and to result into a parametric distribution of the data. Hence, from our previous experience, we were reassured that experiments conducted in triplicate could give representative data.

The paper would benefit from additional discussion on how the findings may influence the selection of dressings for diabetic foot ulcers versus other types of chronic wounds (e.g., venous ulcers).

We have now added to the conclusion section recommendations about the possible criteria of choice of dressings, particularly in the case of diabetic wounds.

---

## [Decision Letter · Decision Letter 1]

26 Dec 2024

A Systematic In Vitro Study of the Effect of Normoglycaemic and Hyperglycaemic Conditions on the Biochemical and Cellular Interactions of Clinically-available Wound Dressings with Different Physicochemical Properties

PONE-D-24-27036R1

Dear Dr. Santin,

We’re pleased to inform you that your manuscript has been judged scientifically suitable for publication and will be formally accepted for publication once it meets all outstanding technical requirements.

Kind regards,

Amitava Mukherjee, ME, Ph.D.

Academic Editor

PLOS ONE

Additional Editor Comments (optional):

Reviewers' comments:

Reviewer's Responses to Questions

**Comments to the Author**

1. If the authors have adequately addressed your comments raised in a previous round of review and you feel that this manuscript is now acceptable for publication, you may indicate that here to bypass the “Comments to the Author” section, enter your conflict of interest statement in the “Confidential to Editor” section, and submit your "Accept" recommendation.

Reviewer #1: All comments have been addressed

2. Is the manuscript technically sound, and do the data support the conclusions?

Reviewer #1: Yes

3. Has the statistical analysis been performed appropriately and rigorously? 

Reviewer #1: Yes

4. Have the authors made all data underlying the findings in their manuscript fully available?

Reviewer #1: Yes

5. Is the manuscript presented in an intelligible fashion and written in standard English?

Reviewer #1: Yes

6. Review Comments to the Author

Reviewer #1: The authors have provided explanations to my previous concerns, and this revised version might be acceptable.

7. PLOS authors have the option to publish the peer review history of their article (what does this mean?). If published, this will include your full peer review and any attached files.

Reviewer #1: No

---

## [Editor Report · Acceptance letter]

15 Jan 2025

PONE-D-24-27036R1 

PLOS ONE

Dear Dr. Santin, 

I'm pleased to inform you that your manuscript has been deemed suitable for publication in PLOS ONE. Congratulations! Your manuscript is now being handed over to our production team.

Kind regards, 

on behalf of

Professor Dr. Amitava Mukherjee 

Academic Editor

PLOS ONE